# Data-driven prediction of complex crystal structures of dense lithium

Xiaoyang Wang[1,2,6], Zhenyu Wang[1,6], Pengyue Gao[1], Chengqian Zhang[3,4], Jian Lv [1] ✉, Han Wang [2,5] ✉, Haifeng Liu[2], Yanchao Wang [1] & Yanming Ma [1] ✉

Lithium (Li) is a prototypical simple metal at ambient conditions, but exhibits remarkable changes in structural and electronic properties under compression. There has been intense debate about the structure of dense Li, and recent experiments offered fresh evidence for yet undetermined crystalline phases near the enigmatic melting minimum region in the pressure-temperature phase diagram of Li. Here, we report on an extensive exploration of the energy landscape of Li using an advanced crystal structure search method combined with a machine-learning approach, which greatly expands the scale of structure search, leading to the prediction of four complex Li crystal structures containing up to 192 atoms in the unit cell that are energetically competitive with known Li structures. These findings provide a viable solution to the observed yet unidentified crystalline phases of Li, and showcase the predictive power of the global structure search method for discovering complex crystal structures in conjunction with accurate machine learning potentials.

Light alkali metals lithium (Li) and sodium (Na) adopt high-symmetry cubic crystal structures and exhibit nearly free-electron behaviors at ambient conditions. These prototypical simple metals, however, undergo drastic property changes under strong compression, showcasing enhanced superconducting critical temperature $T_c$[1,2], metal-semiconductor/insulator transitions[3–6], anomalous melting curves[7–10] and emergence of symmetry-breaking structures[11,12]. These remarkable properties are accompanied by a series of transitions into complex crystal structures in the pressure–temperature (P–T) phase diagram, which presents formidable challenges to both experimental measurements and theoretical elucidation. The ground state of Li was recently found to adopt an *fcc* structure rather than the previously recognized 9*R* structure[13]. Meanwhile, temperature-induced phase transitions around the melting minimum at high pressures were observed in Na[12] and recently proposed for Li[10,14], but determination of pertinent Li structures remains an open question.

Concerted experimental and theoretical efforts in past decades have led to the construction of phase diagram of Li up to ~130 GPa at low temperatures. Its crystal structure transforms from the *bcc* (298 K) to *cl*16 structure at 42 GPa through the intermediate *fcc* and *hR*1 structures[11], followed by a complex phase transition sequence, $cl16 \xrightarrow{\sim 62\,GPa} oC88 \xrightarrow{\sim 70\,GPa} oC40 \xrightarrow{\sim 95\,GPa} oC24$, accompanied by an intriguing metal-semiconductor-metal transition[5,6,8]. The melting curve of Li reaches a maximum followed by a pronounced minimum[8–10], similar to those observed in other alkali metals[7,15–17]. The melting minimum of Li occurs in the pressure range of 40−60 GPa, but the melting temperature ($T_m$) is under debate, as a high P−T experiment for Li is challenging due to the difficulties associated with the sample containment and the degradation of the diamond cell by Li above 20 GPa and 200 K[8–10]. Single crystal X-ray diffraction measurements by Guillaume et al. determined $T_m$ to be 190 K[8]. But later resistivity

[1]Key Laboratory of Material Simulation Methods & Software of Ministry of Education and State Key Laboratory of Superhard Materials, College of Physics, Jilin University, 130012 Changchun, People's Republic of China. [2]Laboratory of Computational Physics, Institute of Applied Physics and Computational Mathematics, Fenghao East Road 2, 100094 Beijing, People's Republic of China. [3]DP Technology, 100080 Beijing, People's Republic of China. [4]College of Engineering, Peking University, 100871 Beijing, People's Republic of China. [5]HEDPS, CAPT, College of Engineering, Peking University, 100871 Beijing, People's Republic of China. [6]These authors contributed equally: Xiaoyang Wang, Zhenyu Wang. ✉e-mail: lvjian@jlu.edu.cn; wang_han@iapcm.ac.cn; mym@jlu.edu.cn

measurements by Schaeffer et al. found a higher $T_m$ of 306 K[9]. The discrepancy was attributed to the possible emergence of a glassy or highly disordered phase below $T_m$. Recently, Frost et al. proposed a rapid compression scheme by taking advantage of high X-ray flux at modern synchrotron, which allows X-ray diffraction data for Li to be collected at higher P–T conditions. They revisited this disputed region and found $T_m$ between 275 and 320 K, below which evidence for yet undetermined crystalline phases was found[10].

Ab initio simulations[18–20] predicted $T_m$s between 250–300 K, in fair agreement with experimental measurements[9,10]. Calculations using the Wigner-Kirkwood approximation for nuclear quantum effects (NQEs) in the liquid and the lattice entropy in the solid[21] produced a $T_m$ of 200 K between 40 and 60 GPa, close to the measured value reported in ref. 8. All these works regard cI16 as the solid phase before melting, and a recent computational study proposed a temperature-induced phase transition from the *cI16* to a *C2/m* structure before melting, predicting a $T_m$ of at least 300 K and a pre-melting regime with collective atomic motions[14]. It is known that Na has crystal structures containing up to 512 atoms in a unit cell near its melting minimum[12]. Given the similarities between Li and Na, along with recent computational and experimental findings[9,10,14], it is reasonable to expect a rich polymorphism and the emergence of complex crystal structures for Li near its melting minimum.

In this work, we explore the potential energy surface (PES) of Li around the melting minimum using the swarm-intelligence-based CALYPSO method[22–25] combined with a machine-learning potential named Deep Potential (DP)[26,27]. This approach allows examination of crystal structures containing up to 200 atoms, which was previously prohibited by the high computational cost of global structure searches based on density functional theory (DFT). Our study identifies four crystal structures denoted by Pearson notations of aP160, oP192, oP48, and tI20. Among them, the aP160 and oP192 structures are of the same kind comprising parallel atomic chains, the oP48 structure is composed of symmetry-related atomic layers analogous to the experimentally identified oC88 and oC40 phases, and the tI20 structures can be seen as a commensurate host-guest structure corresponding to incommensurate composite structures in heavier alkali metals. Gibbs-free energy calculations for various experimentally observed and theoretically predicted Li phases indicate a complex energy landscape with multiple shallow minima, suggesting that the four predicted structures are experimentally accessible as promising candidates for the undetermined phases observed in recent experiments[10]. The present work demonstrates an efficient approach for exploring multi-minima PES and identifying complex crystal structures under diverse P–T conditions, laying the foundation for elucidating broad range of properties.

## Results and discussion
### Crystal structures
Our structure searches used the DP model as the PES calculator[26,27] for the CALYPSO method[22–25], which has successfully predicted the structures of a large number of systems[28], including the semi-conducting *oC*40 phase of Li[5]. Taking advantage of this accelerated structure optimization scheme supported by the efficient DP model, we performed high-throughput searches with simulation cells containing 1–200 atoms at 50 and 60 GPa. More than 600,000 structures were sampled, among which about 5000 lowest-enthalpy structures were further subjected to duplicate elimination and refinement by the DFT calculations. Further computational details for DFT calculations and DP constructions are provided in Supplementary Sec. SI and SII, respectively. Using this approach, we reproduced the experimentally observed cI16 and oC88 phases of Li and found four additional energetically competitive structures denoted by aP160, oP192, oP48, and tI20 shown in Fig. 1. These structures contain large numbers of atoms in the unit cell and complex bonding arrangements. Pertinent crystal structure files are provided in the Supplementary Data file.

The aP160 structure [Fig. 1a] has the *P*1 space group symmetry with 160 atoms in the unit cell and atomic chains arranged in parallel along the *c* axis. Every four chains form a rhombic prism stacked alternatively in the *b*-*c* plane. Several similar structures including the recently proposed dimerized *C2/m* structure[14] also have been predicted by our searches. These structures have smaller unit cells and higher symmetries [Supplementary Sec. SIIIA], and they are different from the aP160 in the arrangement of the prisms and are enthalpically degenerate with the aP160 at 50 and 55 GPa, but become unfavorable at higher or lower pressures (Supplementary Fig. S5). An interesting phenomenon observed in the aP160 structure is that the atomic chains are gradually distorted with increasing pressure, similar to structural changes during melting of a crystal (Supplementary Fig. S6). This phenomenon is not observed in other structures of this type due to the constraints of the higher symmetry and smaller unit cells, and it is responsible for the favorable enthalpy of the aP160 of Li. Another notable phenomenon observed in the aP160, as well as other structures described below, is the wide distribution of distances of the

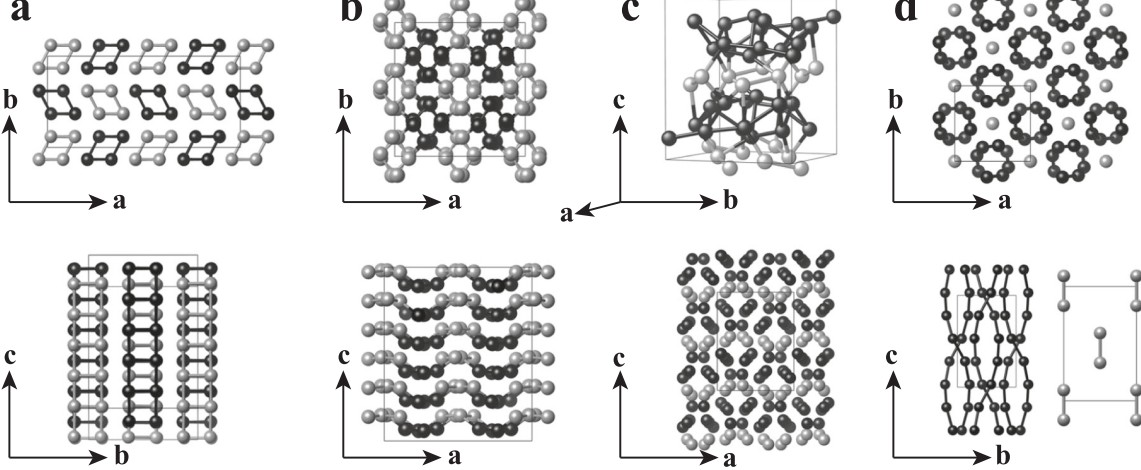

**Fig. 1 | Predicted crystal structures of Li. a** A triclinic structure containing 160 atoms in a unit cell with *P*1 symmetry (Pearson notation aP160), **b** a orthorhombic structure containing 192 atoms in a unit cell with *Pcc*2 symmetry (Pearson notation oP192), **c** a orthorhombic structure containing 48 atoms in a unit cell with *Pbcn* symmetry (Pearson notation oP48), and **d** a tetragonal structure containing 20 atoms in a unit cell with *I*4 symmetry (Pearson notation tI20). Two views are given in each case. Atoms in different rhombic prisms in **a**, **b** or different layers in **c** are shown in black and gray, respectively, while atoms in the host-guest structures in **d** are shown in black and gray, respectively. See text for detailed descriptions.

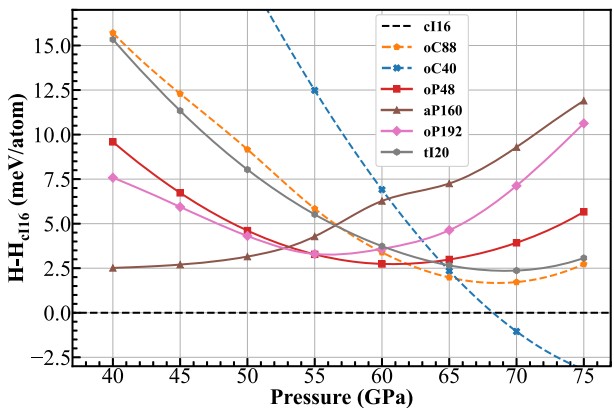

**Fig. 2 | Enthalpies of Li phases.** DFT enthalpy per atom for experimental (dashed line) and predicted structures (solid line) of Li as a function of pressure between 40 and 75 GPa relative to the cI16 phase.

nearest-neighbor contacts (Supplementary Fig. S6). At 50 GPa, the Li...Li distances are between 2.01–2.47 Å. This bond alternation is also found in the experimentally identified oC88 and oC40 phases, and is a common feature of Li at high pressure due to Peierls distortion[5,29,30].

The oP192 structure [Fig. 1b] in the *Pcc2* symmetry contains 192 atoms in the unit cell. It is a aP160-like structure but formed by more distorted rhombic prisms (along the *c* axis), with wrinkled atomic layers in *a-b* plane stacked along the *c* axis.

The oP48 structure [Fig. 1c] shares similar structural features with the oC88 and oC40 phases, exhibiting a 3D irregular network [see top of Fig. 1c] and containing a series of atomic layers [see bottom of Fig. 1c]. The oP48 structure adopts the *Pbcn* symmetry, with six crystallographically distinct Li atoms in the eightfold 8*b* Wyckoff site, forming two symmetry-related 8-atom layers (shown in gray) and two symmetry-related 16-atom layers (shown in black).

The tI20 structure [Fig. 1d] adopts the *I4* symmetry in a commensurate host-guest structure. The host structure comprises double helixes along the *c* axis (shown in black), while the guest structure is composed of Li-Li dimers located at the body center of the tetragonal cell (shown in gray). One of the most intriguing discoveries in alkalis under pressure is the emergence of incommensurate host-guest structures in Na-V, K-III, and Rb-IV[12,31–33]. The tI20 structure of Li is a counterpart of this type of structures. The current method with the periodic boundary conditions is unfeasible to describe an incommensurate phase. However, from our experiences on predicting the structure of solid chlorine[34], our structure search method can provide information for estimating whether there exist incommensurate phases. If an incommensurate structure exists, the method will find several energetically nearly degenerate and geometrically similar structures, which originate from different commensurate approximations of the incommensurate structure. Moreover, these predicted structures would show imaginary phonons because of structural frustration caused by the imposed periodic boundary conditions. Such scenarios do not emerge during the structure search for the tI20 structure, which indicates that it is unlikely for a tI20-like incommensurate structure to exist.

### Enthalpy of various Li phases

We have performed enthalpy calculations using DFT at a high level of accuracy for the four predicted Li structures along with those of the experimentally observed cI16, oC88, and oC40 phases, and the results are plotted as a function of pressure in Fig. 2. Our results for the known Li phases are in good agreement with previous theoretical calculations[6,35]. The cI16 is stable up to ~67 GPa, beyond which the semiconducting oC40 is enthalpically favorable; meanwhile, the oC88 is less favorable than the cI16 at all pressures (about 2 meV higher in

enthalpy than the cI16 at 65 GPa). Experimentally, the oC88 was observed at 77 K above ~62 GPa[8], and a theoretical calculation assessed its Gibbs-free energy under harmonic approximation and found that the free energy of the oC88 at 65 GPa was lower than that of the cI16 at finite temperatures[35]. Similarly, all four predicted structures of Li are less favorable in enthalpy than the cI16 by a few meV. However, these structures show lower enthalpies than the oC88 between 40–62 GPa, and among them the aP160 has the lowest enthalpy below 53 GPa, while the oP192 and oP48 have the lowest enthalpies at 55 and 60 GPa, respectively. The tI20 is nearly degenerate with the oC88 in the entire pressure range considered.

### Dynamic stability

The dynamic stability of the oP48 and tI20 structures was verified by the absence of imaginary frequency in the Brillouin zone via phonon calculations (Supplementary Sec. SI, Fig. S2). For the aP160 and oP192 structures, the large unit cells make DFT-based phonon calculations prohibitively expensive, thus the dynamic stability is examined by Deep-Potential molecular dynamics (DPMD) simulations at $T = 100$ K, using the *NPT* ensemble, at 45 and 55 GPa, respectively. The mean-square displacements remain nearly constant during the 1 ns simulation period, indicating that these two structures are dynamically stable (Supplementary Sec. IIIB, Fig. S7).

### Thermodynamic stability of various Li phases near the melting minimum

The thermodynamic stability of various Li phases near the melting minimum was determined by their relative Gibbs-free energy. We first revisited the relative stability between the experimentally observed cI16 and oC88 phases by calculating the Gibbs-free energy via two approaches: (i) under harmonic approximation using both finite displacement (FD) and density functional perturbation theory (DFPT) methods within DFT (Supplementary Sec. SI), and (ii) considering anharmonicity using thermodynamic integration (TI) through DPMD (Supplementary Sec. SIV). The DP error of the Li-DP-Hyb2 model relative to DFT increases with increasing temperature, but stays less than 1 meV/atom at temperatures below 150 K (Supplementary Sec. SIIC). The statistical uncertainty for molecular dynamics (MD) calculations was estimated to be well below 1 meV/atom.

The calculated Gibbs-free energy as a function of temperature is shown in Fig. 3. All the calculations produced higher free energy of the oC88 than that of the cI16. Under harmonic approximation, the FD and DFPT methods give consistent results with a difference of less than 1 meV/atom. The relative free energy of the oC88 slightly decreases with increasing temperature, but remains above that of the cI16 up to 300 K. At 0 K, the zero-point energies for the cI16 and oC88 calculated by FD (DFPT) are 83.1 (83.0) and 83.2 (83.4) meV/atom, respectively. This result implies that while NQEs contribute significantly to the free energy, they are largely canceled between the two structures, in agreement with previous ab initio path-integral MD studies[14,20]. Considering anharmonicity, there is a different trend of relative free energy of the oC88, which increases rapidly with increasing temperature. This result indicates that anharmonic effects tend to destabilize the oC88 and play an increasingly important role at increasing temperatures. Overall, our present results suggest metastability of the oC88 against the cI16. This result is inconsistent with the experimental observations, i.e., the cI16 would transform to the *o*C88 above ~62 c at low temperatures[4,8,10,36]. On one hand, from a theoretical point of view, two factors should be further considered to resolve this discrepancy: the rigorous treatments of the NQEs and the electronic exchange-correlation functional. Further considerations of these two effects are interesting but computationally demanding, so they are beyond the scope of the current work. On the other hand, in the experiment, transformation of the cI16 into a kinetically protected metastable phase (here oC88 phase) is not excluded.

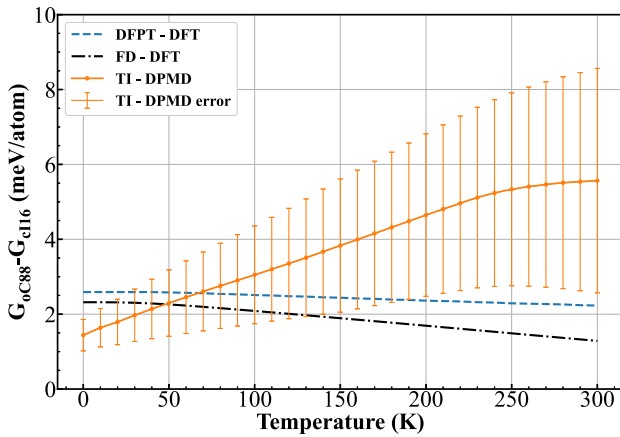

**Fig. 3 | Relative stabilities between oC88 and cI16 phases.** Gibbs-free energy of the oC88 phase as a function of temperature relative to the cI16 phase at 65 GPa, calculated under the harmonic approximation using both finite displacement (FD) and density functional perturbation theory (DFPT) methods through density functional theory (DFT), and considering anharmonicity using thermodynamic integration (TI) through Deep-Potential molecular dynamics (DPMD). The error bar for the TI results is defined as $\sqrt{\epsilon_{model}^2 + \epsilon_{stat}^2}$, where the DP error $\epsilon_{model}$ is the root-mean-square error of the DP predicted energy relative to DFT, $\epsilon_{stat}$ is the statistical uncertainty for molecular dynamics.

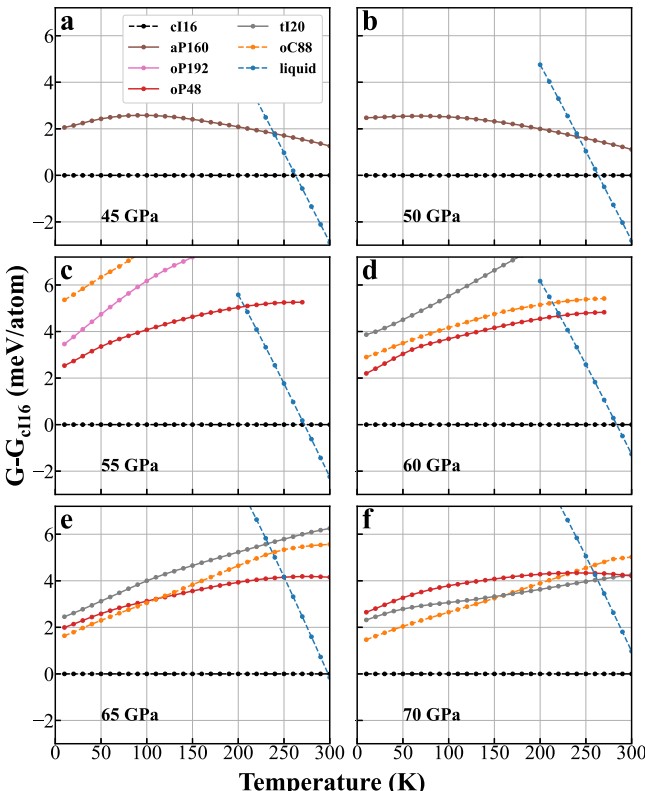

**Fig. 4 | Relative stabilities of Li phases.** Gibbs-free energies of various Li structures relative to the cI16 phase as a function of temperature, calculated using thermodynamic integration (TI) through Deep-Potential molecular dynamics (DPMD) at **a** 45, **b** 50, **c** 55, **d** 60, **e** 65, and **f** 70 GPa.

We now proceed to assess the thermodynamic stability of the four predicted Li structures by calculating their Gibbs-free energy using TI through DPMD. The free energies calculated at 45, 50, 55, 60, 65, and 70 GPa of the predicted, experimental and liquid phases of Li are plotted as a function of temperature up to 300 K and are shown in Fig. 4. The results show that the melting temperature is 262 K at 45 GPa and gradually increases to above 300 K at 70 GPa.

At 45 and 50 GPa [Fig. 4a, b], only the cI16 and aP160 are dynamically stable, and the cI16 is lower in free energy by only ~2–4 meV/atom. Note that the relative free energy of the aP160 gradually decreases with increasing temperature at 45 and 50 GPa. This leads to free energy differences among the cI16, aP160, and liquid phases lower than 2 meV at 250 K. At 55 and 60 GPa [Fig. 4c, d], the predicted oP48 becomes the second most stable structure until melting. The predicted oP192 is dynamically stable only at 55 GPa and shows lower free energies than the oC88. At 65 and 70 GPa [pressures where the oC88 is observed in experiments, Fig. 4e, f], the oC88 is the second most stable structure at low temperatures, while the oP48 and tI20 possess lower free energies than the oC88 above ~110 K (at 65 GPa) and above ~160 K (at 70 GPa), respectively.

The small differences in free energy between various Li phases are sometimes within the errors of the current DPMD simulations, highlighting the difficulty in theoretically assessing the relative stability of Li phases around the melting minimum. For example, the DP error relative to DFT and statistical uncertainty of the DPMD simulations are 1.39 and 0.52 meV/atom at temperatures below 250 K, respectively (Supplementary Sec. IIC). However, several insights can be gained from the present work: (1) The aP160 is the most promising candidate structure at finite temperatures below 50 GPa, as it exhibits nearly degenerated Gibbs-free energy with the cI16 and liquid phases at 50 GPa and 250 K, where all other competing phases are dynamically unstable; (2) the oP48 exhibits superior thermodynamic stability among all the predicted structures and the experimentally observed oC88 at 55 GPa; (3) Li exhibits a nearly flat energy landscape with multiple shallow local minima around the melting minimum, on which the predicted structures compete with the previously reported cI16 and oC88 phases within a small free energy window of only a few meV. These considerations suggest that the four predicted Li structures should be experimentally accessible, and their occurrence would be sensitive to experimental P–T paths.

In summary, we demonstrate in this work that the combination of the state-of-the-art swarm-intelligence-based global optimization and deep learning techniques allows to explore and construct the PES of materials in unprecedented scales and details. By taking the challenging problem of Li phases around the melting minimum as a prominent case study, we discovered four complex crystal structures with large unit cells containing up to 192 atoms that host competing Gibbs-free energies comparable with those previously observed in experiments. Note that the maximum system size in our work is 200 atoms per unit cell, which is one of the largest cell sizes used in the field of structure search to date. However, as a structure of sodium with 512 atoms in a unit cell has been experimentally observed near the melting minimum[12], further exploration of Li systems with larger cell sizes may shed new light on the crystal structure and, therefore, is worth pursuing in future works.

A recent experimental work observed X-ray diffraction peaks of crystalline phases of Li in this P–T region through a rapid compression scheme by taking advantage of high X-ray flux at modern synchrotron[10]. However, the data are not sufficient to unambiguously assign structures due to the difficulties associated with the sample containment and the degradation of the diamond cell. The present work offers strong evidence for the existence of new solid phases of dense Li around the melting minimum, and our findings are expected to stimulate further theoretical and experimental exploration of this intriguing problem.

## Methods
### Structural predictions
The structure searching calculations are performed by using the CALYPSO method[22–25] combined with the DP model as a PES calculator[26,27]. The equipment of DP model rather than DFT allows the

underlying structure optimizations and energy evaluations to be performed very efficiently. By taking advantage of this accelerated searching scheme, high-throughput searches with simulation cells containing 1–200 atoms were performed at 50 and 60 GPa, where a total of more than 600,000 structures are sampled. About 5 000 lowest-enthalpy structures are then subjected to duplicate elimination and refined by the DFT method.

## DFT calculations

The DFT calculations are carried out for the construction of training dataset of DP models, refinement of predicted structures and enthalpy calculations by employing the VASP code[37], which adopts the projector-augmented wave method[38], with $2s^2 2p^1$ treated as valence electrons, and the PBE exchange-correlation functional in the generalized gradient approximation[39,40]. The energy cutoff of 1300 eV and the Monkhorst-Pack k-point sampling grid spacing of $0.1\,\text{Å}^{-1}$ were chosen to ensure the convergence of total energy, forces, and virial tensors below 1 meV, 10 meV/Å and 1 meV/atom, respectively.

Calculations for phonon and Gibbs-free energy under harmonic approximation are carried out using the FD method as implemented in the PHONOPY code[41]. The Hellmann-Feynman forces are calculated using VASP code[37]. The energy cutoff of 600 eV and k-point sampling grid spacing of $0.1\,\text{Å}^{-1}$ were chosen. The Gibbs-free energy under harmonic approximation is also calculated using the DFPT as implemented in the ABINIT code[42], with PBE exchange-correlation functional and energy cutoff of 250 Ry (see Supplementary Sec. I).

## DP constructions

The DP models are trained using highly accurate DFT datasets constructed by the concurrent learning scheme as implemented in the Deep-Potential Generator (DP-GEN)[43]. The DP-GEN[43] is an efficient tool to construct the most compact and adequate dataset for a DP model. Beginning with a preliminary dataset, the DP-GEN runs iteratively through the training, exploration and labeling processes. Two DP models are trained. The first model (Li-DP-Hyb1) is initialized from the known experimental structures of Li and used for accelerating the structure searches. After structure searches, the predicted structures are used to inform the dataset construction of the second model (Li-DP-Hyb2). The Li-DP-Hyb2 model, which contains all information of Li-DP-Hyb1, is used to calculate the Gibbs-free energy by TI through DPMD.

In the training process, four models are trained using the same training dataset and hyper-parameters but different random seeds. The three-body embedding descriptor hybridized with the smooth edition DP descriptor is utilized. The sizes of the 3 hidden layers of the two-body embedding net, three-body embedding net, and the fitting net are (20, 40, 80), (4, 8, 16), and (240, 240, 240), respectively. In the exploration process, the configuration space of Li is sampled through NPT DPMD simulations. The LAMMPS package[44] compiled with the DeePMD-kit[45] support is employed to perform the DPMD[26] simulations. During the DPMD simulations, configurations are calculated by the four DP models obtained in the training process. If the maximal deviation of atomic forces of a configuration lies in between an upper bound and a lower bound, the configuration is considered to be a candidate, whose labels (i.e., the energy, force, and virial tensor) will be calculated by DFT in the labeling processes and included into the training dataset. More technical details for the DP dataset construction and model training are presented in the Supplementary Sec. II.

## Gibbs-free energy calculations using TI through DPMD

The calculation of Gibbs-free energy of a Li phase takes a two-step process. First, we calculate the Gibbs-free energy of a given Li phase relative to a reference system, whose free energy is known. This is realized by the Hamiltonian thermodynamic integration (HTI) at a reference thermodynamic condition, $(T_0, P_0)$, where the dynamic stability of the Li phase should be guaranteed. Then, the free energy difference of the Li phase between the target thermodynamic condition $(T_1, P_1)$ and the reference condition $(T_0, P_0)$ is calculated by the TI along an isobar $(P_0 = P_1)$. The principles of HTI and TI are introduced in ref. 46. The HTI and TI are implemented with the workflow in Deep-Potential thermodynamic integration (DPTI) code, which automatically generates the scripts for DPMD simulations conducted by LAMMPS[44], and manages the pre-processes, post-processes and numeric integrations. For all DPMD simulations, the timestep was set to 1 fs. Before HTI, a NPT DPMD simulation is performed to obtain the proper simulation box at $(P_0, T_0)$. Then a NVT DPMD simulation is performed at $T_0$ to check whether the obtained simulation box is reasonable. The NPT DPMD simulation runs up to 1 ns, and the NVT DPMD simulation runs up to 200 ps. More details for free energy calculation are presented in the Supplementary Sec. IV.

## Data availability

The authors declare that the main data supporting the findings of this study are contained within the paper and its associated Supplementary Information. Source data are provided with this paper.

## Code availability

CALYPSO code is free for academic use, by registering at http://www.calypso.cn. LAMMPS, DP-GEN, DeepMD-kit and DPTI are free and open source codes available at https://lammps.sandia.gov and https://deepmodeling.com, respectively. The other scripts are available from the authors upon request.

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

## Acknowledgements

J.L. is grateful to Prof. G. Csanyi for valuable discussions on construction of machine-learning potentials. The work of J.L. is supported by National Natural Science Foundation of China (Grants No. 12034009, 91961204, 11974134, 52288102 and 11904129). The work of H.W. is supported by the National Science Foundation of China (Grant No.11871110 and 12122103).

## Author contributions

J.L., H.W., and Y.M. designed the research; X.W., and Z.W. performed the calculations; X.W., Z.W., P.G., C.Z., J.L., H.W., H.L., Y.W., and Y.M. analyzed and interpreted the data, and contributed to the writing of the paper. X.W. and Z.W. contributed equally to this work.

## Competing interests

The authors declare no competing interests.
