## [Peer Review File · Nature Communications]

Data-Driven Prediction of Complex Crystal Structures of Dense LithiumREVIEWER COMMENTS

Reviewer #1 (Remarks to the Author):

This is a technically amazingly comprehensive paper using state of the art simulation methods. In this way the authors could discover novel and interesting phases of lithium. The fact that a machine learned potential was used allowed them to treat very large systems that are not accessible to DFT calculations. In summary, this is a very high quality MS in every respect and I recommend publication in its present form.

Reviewer #2 (Remarks to the Author):

Review comments to "Data-Driven Prediction of Complex Crystal Structures of Dense Lithium" by Professor Lv and colleagues. Overall, the paper is quite well written and presented. There are a few points that would need additional discussions and comments before it can be accepted for Nature Communications. As a caveat, we want to point out that we are not experts in the computational methods and would defer to other reviewers with regard to those.

1. Page 1 paragraph 2: "The melting minimum of Li occurs in the pressure range of 40-60 GPa, but the melting temperature is sensitive to the pre-melting crystal structure and nuclear quantum effects." This needs a citation. The melting temperature is in agreement between Frost and Schaffer, and Guillaume is not consistent with these and claims melting where Frost observes clear diffraction from hR1 lithium. Lithium is chemically simple and we find it unlikely that the melting point can depend on the precursor phase. The authors should add comments and discussions on this point.
2. The authors should comment on the possibility of structures with more than 200 atoms/ unit cell, the maximum cell size they studied. For example sodium has a structure with 512 near its melting minima (ref 12 in manuscript). In a similar vein they point out similarities between their commensurate host-guest tI20 phase and the incommensurate (and guest disordered) phases in other alkali metals. It is unclear whether their method is capable of predicting incommensurate phases and what the implications of this limitation are.
3. Last sentence in page 2, start of page 3: They say that their method predicted the recently proposed C2/m structure (ref 14 in manuscript), but make no further mention of it. How does its enthalpy and Gibbs free energy compare to the other structures investigated? If it is unfavorable, comment on the discrepancy with ref 14.
4. Page 4, last paragraph: "The small differences in free energy between various Li phases are sometimes within the errors of current DPMD simulations" . These errors should be quantified.
5. Perhaps most importantly the study finds that cI16 is the only stable solid phase from 45 to 70 GPa. This disagrees with experimental observations (manuscript refs [4, 8, 10] and Matsuoka et al. "Pressure-induced reentrant metallic phase in lithium" <https://journals.aps.org/prb/abstract/10.1103/PhysRevB.89.144103>). Does this expose a weakness in the computational method, or do the authors propose that the oC88 phase is metastable to cI16, in which case what drives its formation from cI16 in experiment? Further commentary to contextualize this result with the existing literature is required.

Responses to Reviewer Reports

Reviewer 1:

This is a technically amazingly comprehensive paper using state of the art simulation methods. In this way the authors could discover novel and interesting phases of lithium. The fact that a machine learned potential was used allowed them to treat very large systems that are not accessible to DFT calculations. In summary, this is a very high quality MS in every respect and I recommend publication in its present form.

Our response:

We thank the review for spending time and effort to evaluate our manuscript and appreciate the reviewer's positive assessment of our reported work.

Reviewer 2:

Review comments to "Data-Driven Prediction of Complex Crystal Structures of Dense Lithium" by Professor Lv and colleagues. Overall, the paper is quite well written and presented. There are a few points that would need additional discussions and comments before it can be accepted for Nature Communications. As a caveat, we want to point out that we are not experts in the computational methods and would defer to other reviewers with regard to those.

Our response:

We thank the reviewers for spending time and effort to evaluate our manuscript and appreciate their positive assessment of our reported work and their insightful suggestions for improving our manuscript. Our point-by-point responses are given below.

1. Page 1 paragraph 2: *"The melting minimum of Li occurs in the pressure range of 40-60 GPa, but the melting temperature is sensitive to the pre-melting crystal structure and nuclear quantum effects." This needs a citation. The melting temperature is in agreement between Frost and Schaffer, and Guillaume is not consistent with these and claims melting where Frost observes clear diffraction from hR1 lithium. Lithium is chemically simple and we find it unlikely that the melting point can depend on the precursor phase. The authors should add comments and discussions on this point.*

Our response:

We thank the reviewers for raising these issues. Following their comments and suggestions, we have rewritten pertinent discussions in the revised manuscript (2nd paragraph on page 1).

2. *The authors should comment on the possibility of structures with more than 200 atoms/unit cell, the maximum cell size they studied. For example, sodium has a structure with 512 near its melting minima (ref 12 in manuscript). In a similar vein they point out similarities between their commensurate host-guest tI20 phase and the incommensurate (and guest disordered) phases in other alkali metals. It is unclear whether their method is capable of predicting incommensurate phases and what the implications of this limitation are.*

Our response:

We thank the reviewers for these constructive comments. The maximum system size in our work is 200 atoms/unit cell, which is one of the largest cell sizes used in the field of structure search to date. This choice is limited by the computational resources. Further exploration of systems with larger cell sizes may shed new light on the crystal structure and, therefore, is worth pursuing when higher computing power becomes accessible. To address the reviewers' comments, we have added discussions in the revised manuscript (3rd paragraph on page 5).

In our reported calculations, we employed the periodic boundary conditions to simulate an infinite solid. It is thus unfeasible to describe an incommensurate phase. However, from our experiences on predicting the structure of solid chlorine [J. Chem. Phys. 137, 064502 (2012)], our structure search method can provide information for estimating whether there exist incommensurate phases. If an incommensurate structure exists, the method will find several energetically nearly degenerate and geometrically similar structures, which originate from different commensurate approximations of the incommensurate structure. Moreover, these predicted structures would show imaginary phonons because of structural frustration caused by the imposed periodic boundary conditions. For lithium, such scenarios do not emerge during the structure search for the *tI20* structure, which indicates that it is unlikely for a *tI20*-like incommensurate structure to exist. In response to the reviewers' comments, we have added discussion on this topic in the revised manuscript (last paragraph on page 2 and 1st paragraph on page 3).

3. Last sentence in page 2, start of page 3: They say that their method predicted the recently proposed C2/m structure (ref 14 in manuscript), but make no further mention of it. How does its enthalpy and Gibbs free energy compare to the other structures investigated? If it is unfavorable, comment on the discrepancy with ref 14.

Our response:

We thank the reviewers for raising this question. The enthalpies of various *mP160*-like structures including the *C2/m* structure were previously shown in Supplementary Sec. III A, Fig. S5(c). These structures show nearly degenerate enthalpies with the *mP160* structure at 50 and 55 GPa, but become unfavorable at higher or lower pressures. We have revised the related description in the 4th paragraph on Page 2. Following the reviewers' suggestion, we further calculated the Gibbs free energy of the *C2/m* structure at 45 GPa compared with those of the *mP160*, *cI16* and liquid phases; the results are shown in Supplementary Sec. III A, Fig. S5(d).

FIG. S5. (a) $C2/m$ and (b) $mP160$ structures of Li viewed along the c axis, with the atoms in different rhombic prisms shown in black and grey, respectively. (c) DFT-calculated relative enthalpies per atom of the $mP160$ and similar structures as a function of pressure between 40 and 75 GPa relative to the $cI16$ phase. (d) The Gibbs free energies of the $C2/m$, $mP160$ and liquid phases relative to the $cI16$ phase versus temperature at 45 GPa using TI via DPMD.

Our results show that the $mP160$ and the $C2/m$ structures are almost energetically degenerate at 0-300 K and both are less favorable in free energy than the $cI16$ phase. This contradicts the theoretical results reported in Ref. 14, where the $C2/m$ structure is found to be more stable than the $cI16$ structure (by less than 1 meV/atom) at temperatures above ~ 200 K. This discrepancy can be attributed to the different choices of parameters in the DFT calculations. Both works use the same DFT code, but our current work adopts a much more accurate set of parameters to ensure better numerical convergence. Specifically, an energy cutoff of 1300 eV and a k-grid spacing of 0.1 \AA^{-1} are used in this work, while an energy cutoff of 650 eV and k-grid spacing of $\sim 0.3 \text{ \AA}^{-1}$ were used in Ref. 14. The results of convergence tests of calculated energy by using larger energy cutoffs and smaller k-grid spacings are shown in Supplementary Fig. S1. We find that the parameters used in Ref. 14 would lead to energy errors as high as a few meV/atom. We have added related discussion in Supplementary Sec. III A.

FIG. S1. Convergence tests on total energy in VASP calculations for the $cI16$ and $C2/m$ structures. Total energies as a function of (a) ENCUT and (b) KSPACING, relative to the results obtained using more stringent parameters (ENCUT = 1500 eV and KSPACING = 0.08 \AA^{-1}).

4. Page 4, last paragraph: "The small differences in free energy between various Li phases are sometimes within the errors of current DPMD simulations". These errors should be quantified.

Our response:

We thank the reviewers for this good suggestion. A detailed description on the accuracy of the Deep Potential model is presented in Supplementary Sec. II C. Following the reviewers' suggestion, we have added pertinent description in the 2nd paragraph on page 5.

5. Perhaps most importantly the study finds that *cI16* is the only stable solid phase from 45 to 70 GPa. This disagrees with experimental observations (manuscript refs [4, 8, 10] and Matsuoka *et al.* "Pressure-induced reentrant metallic phase in lithium" <https://journals.aps.org/prb/abstract/10.1103/PhysRevB.89.144103>). Does this expose a weakness in the computational method, or do the authors propose that the *oC88* phase is metastable to *cI16*, in which case what drives its formation from *cI16* in experiment? Further commentary to contextualize this result with the existing literature is required.

Siegfried Glenzer and Mungo Frost

Our response:

We thank the reviewers for raising this important point. The metastability of the *oC88* phase relative to the *cI16* phase revealed by the DFT-based calculations is one of the interesting findings in this work. This is why in Fig. 3 we show the relative stability between the two phases with different methods/codes to ensure consistent results. However, as the reviewers pointed out, this result is inconsistent with the experimental observations, i.e., the *cI16* phase would transform to the *oC88* phase above ~62 GPa at low temperatures. On one hand, from a theoretical point of view, two factors should be further considered to resolve this discrepancy: the rigorous treatments of the nuclear quantum effects and the electronic exchange-correlation functional. Further considerations of these two effects are interesting but computationally demanding, so they are beyond the scope of the current work. On the other hand, in the experiment, transformation of the *cI16* phase into a kinetically protected metastable phase (here *oC88* phase) is not excluded. Following the reviewers' suggestions, we have added discussions in the 3rd paragraph on page 4.

REVIEWERS' COMMENTS

Reviewer #2 (Remarks to the Author):

The authors have answered all questions and provided the information in the main manuscript and the supplement. We strongly recommend publication of this work in *Nature Communications*.

Reviewer #3 (Remarks to the Author):

The authors have fully addressed all my comments. I recommend this manuscript for publication.

Manuscript ID: NCOMMS-22-50476A

Title: "Data-Driven Prediction of Complex Crystal Structures of Dense Lithium"

Author(s): Xiaoyang, Wang; Zhenyu, Wang; Pengyue, Gao; Chengqian, Zhang, Jian, Lv;
Han, Wang; Haifeng, Liu; Yanchao, Wang; Yanming, Ma

Dear Dr. Manel Mondelo-Martell,

Thank you for your communication concerning our manuscript (ID: NCOMMS-22-50476A) and the enclosed reviewer reports. We are very pleased to see that all the reviewers recommend publication of this work in Nature Communications. We have carefully revised the manuscript according to the editorial requests. All the major changes in the revised manuscript are highlighted in red for a convenient reading, and our point-by-point responses to the reviewer comments are given below.

We hope that our revised manuscript is now suitable for publication in Nature Communications, and we thank you for handling our manuscript and look forward to hearing from you again.

With best regards,

Jian Lv, Han Wang and Yanming Ma
on behalf of all authors

Responses to Reviewer Reports

Reviewer 2:

The authors have answered all questions and provided the information in the main manuscript and the supplement. We strongly recommend publication of this work in Nature Communications.

Our response:

We thank the review for spending time and effort to evaluate our manuscript and appreciate the reviewer's positive assessment of our reported work.

Reviewer 3:

The authors have fully addressed all my comments. I recommend this manuscript for publication.

Our response:

We thank the review for spending time and effort to evaluate our manuscript and appreciate the reviewer's positive assessment of our reported work.